# Healing Process, Pain, and Health-Related Quality of Life in Patients with Venous Leg Ulcers Treated with Fish Collagen Gel: A 12-Week Randomized Single-Center Study

**DOI:** 10.3390/ijerph19127108

**Published:** 2022-06-09

**Authors:** Paulina Mościcka, Justyna Cwajda-Białasik, Maria Teresa Szewczyk, Arkadiusz Jawień

**Affiliations:** 1Department of Perioperative Nursing, Department of Surgical Nursing and Chronic Wound Care, Collegium Medicum in Bydgoszcz, Nicolaus Copernicus University in Torun, 85-821 Bydgoszcz, Poland; jcwajda-bialasik@cm.umk.pl (J.C.-B.); mszewczyk@cm.umk.pl (M.T.S.); 2Department of Vascular Surgery and Angiology, Collegium Medicum in Bydgoszcz, Nicolaus Copernicus University in Torun, 85-094 Bydgoszcz, Poland; ajawien@cm.umk.pl

**Keywords:** venous leg ulcers, fish collagen, health-related quality of life, Skindex-29, CIVIQ

## Abstract

The aim of the study was to assess the effectiveness of fish skin collagen and its impact on healing, pain intensity, and quality of life in patients with venous leg ulcers (VLUs). This study included 100 adults with VLUs. Eligible patients were randomized to either tropocollagen gel treatment (group A, *n* = 47) or placebo alone (group B, *n* = 45). We applied the gel to the periwound skin for 12 weeks. All groups received standard wound care, including class 2 compression therapy and wound hygiene procedures. We assessed the healing rate (cm^2^/week) and quality of life (QoL) using the Skindex-29 and CIVIQ scales. In group A, more ulcers healed, and the healing rate was faster. In both study groups, patients showed a significant improvement in quality of life after the intervention, but there was a greater improvement in the tropocollagen group. In group A, the greatest improvement was related to physical symptoms and the pain dimension. This study showed that the application of fish collagen gel to the periwound skin improves the healing process and QoL in patients with VLUs. The 12-week treatment with collagen reduced the severity of physical complaints, pain, and local skin symptoms, which determined the quality of life in patients with VLUs to the greatest extent.

## 1. Introduction

Venous leg ulcers (VLUs) are the most common chronic wounds of the lower limbs, which constitute approximately 70–80% of all leg ulcers [1]. Currently, VLUs are considered a problem worldwide. In the industrialized countries, 1–3% of the adult population are affected by active or healed ulcers [2,3], but among the population aged 65 and above, the incidence of VLUs rises to 5% [1,4,5]. Venous hypertension resulting from venous reflux or obstruction is believed to be the main underlying mechanism of VLUs [3]. At present, compression therapy, modern topical treatment, and wound care constitute the mainstay of treatment for patients without coexisting blood pressure disorders. In some cases, venous ablation and surgical intervention to correct superficial venous reflux [2,3,5] may be beneficial for the healing process. Comprehensive therapy also takes other aspects into account, such as the hygiene of the periwound skin and the entire limb, physiotherapy, and exercises which improve calf muscle pump function, pain management, and nutritional support, among others [1,2,3,4,5,6,7,8]. Venous ulcers usually heal within 6 months, if well managed [1]. Unfortunately, even up to 15–30% of chronic VLUs may not respond to standard treatment and will remain unhealed, even after a 12-month intervention [1,9,10,11]. A high recurrence rate may also be observed, according to various authors [1,11,12,13], from 30 to even 70% after 6 months.

Likewise, as in the case of other chronic diseases, long-term and recurring clinical conditions put the patients at risk for many negative consequences and great physical, mental, and emotional stress, which has been demonstrated in multiple studies [14,15,16,17,18,19,20]. Ulcers are accompanied by pain and mobility restrictions [16,17,18,19]. As a result of the physical effects of the disease, or self-imposed isolation, the ability to fulfill one’s social and professional roles decreases. Low mood and depression may ensue [21]. The costs of long-term therapy and the social consequences of the disease increase, including absenteeism and disability. Thus, comprehensive care of the patient with a chronic wound must not be limited to the wound only. The substantial impact the disease has on patients’ daily life and functioning needs to be considered. While looking for new therapies which improve healing, we also expect these therapies to improve persistent symptoms, such as pain, itch, excessive dryness, skin scaling, eczema (stasis/contact/allergic dermatitis), or the prolonged localized inflammation causing them. It has been demonstrated that multiple cellular, molecular, and biochemical disorders that delay healing occur in chronic wounds, including VLUs. Overexpressed inflammatory pathways were observed, mainly sustained by high levels of neutrophil elastase, matrix metalloproteinases (MMPs), urokinase-type plasminogen activator (uPA), and extracellular MMP inducer (EMMPRIN and CD147), and decreased activity of the tissue inhibitors of MMPs (TIMPs) and consequently, the degradation of the extracellular matrix (EMC) [1]. In addition, activation of leukocyte activity (macrophage (MP), mast cells (MC), and T-lymphocytes (TL)) occurs. Leukocytes express a variety of cytokines (with both direct and indirect effects), causing a continuous pro-inflammatory and inflammatory environment [1]. Disruption in keratinocyte and fibroblast activity, as well as inadequate or incorrect collagen production—one of the most important proteins regulating the healing process—are typical. Collagen provides structural support to the skin (scaffolding), but also acts as a key signaling molecule for ECM [22,23]. Its optimal amount, structure, and function are crucial for the reconstruction of a stable skin barrier and wound closure. The results of in vitro research suggest that topical collagen supplementation may modulate the chronic wound environment and have a positive effect on healing [22,23,24]. This has been confirmed by the collagen dressings used for many years, which are biological and bioactive, containing hemostatic properties and an additional antimicrobial agent [24,25,26,27]. Gould, L.J. [23] reports that the triple helix structure of native collagen is likely to be the most optimal metalloprotease substrate, with angiogenic and chemoattractant properties, but it is rapidly degraded under physiologic conditions. The degradation rate and mechanical properties can be manipulated via cross-linking and sterilization methods. In the production process, it is recommended to use methods that stabilize the structure of the material and at the same time, allow for an increase in the efficiency of chemical cross-linking without reducing the biological efficiency. Unfortunately, some of the methods of collagen production can be expensive and unprofitable, resulting in a limited use of collagen biomaterials in the treatment of chronic wounds [23]. Fish collagen has a relatively low molecular weight (lower intermolecular cross-linking), which makes it possible to obtain a fully biologically active molecule in the form of a helix [28], so we used it.

We used a fish collagen gel intended for application on the periwound skin, not directly into the wound bed. We have assumed that the improvement of periwound skin is as important as the wound microenvironment, since epithelialization starts with the healthy wound edges. In preliminary studies, we have shown that the collagen gel applied in accordance with the adopted protocol penetrates well into the stratum corneum and improves the overall skin condition [29,30,31]. Infrared thermography showed that the 12-week treatment with collagen reduced the severity of the local inflammatory reaction compared to the control group [32]. In this part of the study, we assessed the effectiveness of fish skin collagen and its impact on healing, pain intensity, and quality of life in VLU patients.

## 2. Materials and Methods

### 2.1. Participants

This study included 100 adults (>18 years) with chronic venous ulcers. Recruitment occurred at highly specialized national center for chronic wound healing, in the period from 2016 to 2019. The study classification criteria included the presence of a leg ulcer (area between 5 and 50 cm^2^), CVI, as proven by scanning the lower extremity blood vessels (duplex scan), a duration of ulceration >3 months, an ankle brachial index (ABI) of 0.9–1.3, and a lack of clinical symptoms of infection. The exclusion criteria were ulcerations of mixed etiology, other than venous or undiagnosed, and coexisting lower limb disorders. Eligible patients were randomized to either tropocollagen gel treatment (group A) or placebo alone (group B) using computer generated random numbers. We included only 92 patients with complete measures and correctly completed QoL questionnaires (A, *n* = 47; B, *n* = 45) in the final statistical analysis (Figure 1). The characteristics of the patients included in the analysis are presented in Table 1.

### 2.2. Interventions

All groups received standard wound care twice weekly for 12 weeks, or until healing was complete. Standard treatment for VLUs included class 2 compression therapy (40 mmHg) with Matopress short-stretch bandages and Matosoft Natural cotton wool pads (TZMO Matopat, Toruń, Poland), used for skin protection purposes. Wound hygiene procedures [33] included cleansing the wound and periwound skin, debridement, and renewing the wound edges, using the dressing appropriate to the healing phase. Moreover, patients in group A had tropocollagen gel applied on periwound skin (we administered type I collagen), while patients in group B received a placebo. We applied 5 cm^3^ of the formulation at a time (that is, 10 pumps of the dispenser). The gel was applied on clean skin within 2 cm from the wound edges. The gel was massaged into the skin moistened with saline solution (approximately 10 mL) using circular finger motions (while wearing disposable gloves). Halfway through the application process, the skin was moistened with saline (0.9% solution) and gently rubbed in the remaining gel that had not been fully absorbed. The total application time was equal to 15 min. The application took place twice a day for 12 weeks (even if the wound had completely healed). The nurse applied the gel during appointments at the clinic, while patients applied it on their own at home. The patients had been instructed on how to take care of the wound earlier, and each bottle of gel came with a written instructions for the application process. In this study, we used type I collagen from the skin of silver carp (Collagen Active Science Sp. z o.o., Poznań, Poland), which we previously described in the article on the effectiveness of collagen on the healing process of VLUs [29,30,31,32,34]. The collagen in the gel was in its native form; the denaturation temperature was found at 33.4 °C (by the viscometric method). The gel contained water, lactic acid, preservative (Rocoal, M.D.), and collagen at about a 1% concentration. Electrophoretic studies showed the presence of 200 kDa collagen β-chains and 130 kDa collagen α-chains.

### 2.3. Ulcer Area

We assessed the dynamics of wound healing by planimetry every two weeks during a 12-week study. Moreover, at week 24 since the beginning of the study, the patients were invited to a single follow-up appointment, during which all the measurements were taken once again. Wound area (cm^2^) calculations were performed using a Visitrak digital wound measuring device (Visitrak Digital: Digital-Pad, Smith & Nephew, Schwechat Austria). The primary endpoint, or complete wound healing, was defined as a 100% reduction in wound area. We also assessed health-related quality of life and pain intensity.

### 2.4. Quality of Life

To assess the quality of life, we used standardized tools—the Polish version of the Skindex-29 and the Chronic Venous Insufficiency Questionnaire (CIVIQ) scales.

The Skindex questionnaire is an instrument intended for the assessment of the quality of life in individuals with dermatological disorders and was developed by Chren, M. et al. [35,36]. We used the Polish version of the Skindex-29 questionnaire, adjusted by Janowski, K. [37]. The studies showed that the split-half reliability and diagnostic accuracy of Skindex-29 are high, and this tool can be used to assess the quality of life of patients with vascular leg ulcers [38]. The Skindex-29 includes 29 items grouped into three subscales: (A) physical symptoms involving the skin, (B) psychosocial functioning, including everyday activities, role functioning, and social contacts, and (C) the emotional sphere. The respondents choose an answer corresponding to the frequency (never, rarely, sometimes, frequently, all the time) with which they have experienced any of the problems during the last month. The answers are scored between 1 and 5 points, respectively. The points are summed, and the quality of life score is obtained, ranging between 29 points (highest quality of life—the lack of negative effects of the condition) and 145 points (the worst quality—maximum negative influence of the condition). The scores can also be calculated for each subscale.

The Chronic Venous Insufficiency Questionnaire (CIVIQ) was developed by Prof Launois, with an educational grant from SERVIER [39,40]. We used the Polish language version of CIVIQ-20 from the official CIVIQ website. The questionnaire has 20 items and four dimensions, including pain (4 items), physical dimension (4 items), psychological dimension (9 items), and social dimension (3 items), all scored on the Likert scale from 1 (no symptom, sensation, or trouble) to 5 (highest intensity or frequency, depending on the item). The value of individual domains is calculated based on the sum of the items. The overall result of all items is the so-called the Global Index Score (GIS), which was calculated according to the authors’ instructions: GIS = ([Final score − 20]/80) × 100. The lower the value of each domain/GIS, the higher the quality of life [41].

Health-related quality of life was assessed 3 times: at the beginning of the treatment (initial assessment), at the end of the treatment (12 weeks later), and at the follow-up visit (24 weeks later).

To assess the improvement in the individual domains of the quality of life scales, we calculated the percentage of increase from the baseline.

We used an 11-point numeric rating scale (NRS 11) to assess pain intensity—with 0 representing one pain extreme (e.g., “no pain”) and 10 representing the other pain extreme (e.g., “worst pain imaginable” and “pain as bad you can imagine”). We assessed pain intensity every 2 weeks, along with the wound assessment and follow-up after 24 weeks [42].

### 2.5. Ethics Statement

The project received ethical clearance as a prerequisite for approval for funding from the National Centre of Research and Development as a part of the Applied Research Projects (No NCBiR, PBS3/B7/28/2015). The study protocol was approved by the local bioethics committee of the Nicolaus Copernicus University in Torun, and Ludwik Rydygier Collegium Medicum in Bydgoszcz (No KB 69/2015), and the study was conducted in accordance with the Helsinki Declaration of 1975. All enrolled patients signed a written informed consent form. Patient’s data were managed in accordance with the Polish Data Protection Act [43].

## 3. Results

In group A, more patients with ulcers healed (27.7% vs. 24.4% after 12 weeks and 53.2% vs. 40% after 24 weeks), and the healing rate was also faster, but there were no statistically significant differences between the groups.

There were no significant differences between the groups in the median total Skindex-29 and CIVIQ scores at both baseline and weeks 12 and 24, but in both study groups, patients showed a significant improvement in quality of life after the intervention. A greater improvement in the quality of life was shown in the tropocollagen group. The increase in the overall quality of life, according to the Skindex-29, in groups A and B was 17% vs. 7.75%, from baseline after 12 weeks, and 21.3% vs. 15.5% after 24 weeks. The increase in the overall quality of life according to the CIVIQ in groups A and B was 17.6% vs. 9.1%, from baseline after 12 weeks, and 23.3% vs. 18.27% after 24 weeks (Table 2).

In group A, the greatest improvement in the quality of life on the Skindex-29 scale was related to physical symptoms (compared to group B, the score increased by 16.7% vs. 3.4% after 12 weeks and by 24.3% vs. 15.3% after 24 weeks). In group B, the greater increase in the score concerned the emotional sphere, but the difference was not significant. The improvement in the quality of life in the psychosocial functioning of both groups was comparable. When analyzing the individual items on the scale, the greatest improvement was observed for “skin soreness,” “skin burning, stinging,” “sleep quality,” and “skin sensitivity” (in all, significantly greater improvement in group A, *p* < 0.05). Moreover, in group A, there was a significantly greater improvement (*p* < 0.05) related to “hobby and professional work,” and in group B, “attitudes of loved ones” and “despondency” (*p* < 0.05). There was no significant improvement in either of the two groups in the items related to “fear of skin deterioration,” “worry about leaving scars,” and “embarrassment” (Table 3).

In the CIVIQ scale, the greatest improvement was seen in the pain dimension—significant changes in scores were noted in both groups. In group A, there was a greater improvement in psychological dimension (compared to group B, by 15.8% vs. 6.2% after 12 weeks, and by 18.3% vs. 10.3% after 24 weeks). In the other dimensions, the changes in the scores of both groups were comparable. Analyzing the individual statements according to the scale, we showed that in group A, there was a greater decrease in the intensity of pain in the ankles and legs. The improvement in pain-related sleep quality was also high, but comparable. In group A, there was a significantly greater improvement in the response to the statements, “I have become tired quickly,” and “I have become irritated easily” (*p* < 0.05). There was no improvement in items 11 and 7 in any of the groups, and only slight improvement in item 6 (all related to physical activity), despite their significant impact on overall quality of life (Table 4).

In both groups, changes in the quality of life were significantly correlated with the progress of the wound healing process (with a very strong positive correlation for both the Skindex-19 and CIVIQ scales) (Figure 2a,b).

The mean pain intensity decreased significantly during the intervention; after 12 weeks, it decreased by 3.07 points in group A, and by 2.49 points in group B. In the following 12 weeks, the change in average pain intensity occurred only in group A (by another 0.13 points). There were no intergroup differences in pain intensity, either at the baseline or during the intervention. Pain intensity was poorly correlated with the initial wound surface (negligible positive correlation in group A, weak in group B, both statistically significant) and strongly correlated with the wound healing process (very strongly positive in group A, strongly in group B, both statistically significant) (Figure 3).

## 4. Discussion

Collagen is the most common protein, with a wide range of biomedical applications. This modern biomaterial can be obtained from bovine skin and tendons, porcine skin, intestine, or bladder mucosa, and even rat tail. Alternatively, collagen can be produced by heterologous expression in mammalian, insect, and yeast cells. It can also be produced by *Escherichia coli* [44]. Due to its low antigenicity and inherent biocompatibility with most endogenous tissue, natural collagen may be used for many pre- and post-operative surgical procedures (e.g., as adhesives), and also as a wound dressing. It is especially dedicated to chronic wounds, including VLUs, with a disturbed microenvironment and excessive inflammation that inhibits healing. Collagen-based wound dressings (including partially purified skin, hydrolyzed collagen and collagen sponge, fiber, powder or composite dressings) have been shown to have practical and economic advantages over growth-factor and cell-based dressings in the treatment of full-thickness wounds [45]. Several studies have shown the advantage of collagen dressings compared to the standard of care [46,47,48,49] and alginate dressings [45,50]. Adding a collagen dressing to the standard of care protocol has been shown to increase the probability of ulcer healing from 0.11 to 0.49 by 6 months and may reduce management costs by 40% [49]. In the treatment of chronic wounds, it is reported that the triple-helix native collagen is the most desirable form [23,45]. It has strong angiogenic and chemotactic properties [23]. We used this type of collagen—a fully biologically active molecule in the form of a helix—obtained from the skin of silver carp. It is now more and more often emphasized that fish tropocollagen is a novel product of rich source and high biocompatibility, and unlike mammalian collagen, it has low risk of virus transmission and low biological risk [24,28,33,51,52]. One study used omega-3-rich fish skin grafts to treat healing-resistant diabetic foot ulcers (DFUs). Compared to the standard of care, there was a significant reduction in the percentage of wound area (72.8% vs. 41.2%) and a greater number of healings (67% vs. 32%) after 12 weeks of treatment [53].

We applied tropocollagen gel to the periwound skin, not the wound bed. We assumed that the improvement of the skin condition and its parameters will increase the wound’s potential for effective healing. Previously, we observed that the gel applied to the skin around the ulcer first improved the general condition of the skin [28,29,30], and then accelerated the healing rate (significantly only from the 8th week of use, *p* < 0.05). This was manifested by a reduction in the local temperature around and in the wound area (in the IRT) [32,54], as well as a reduction in the severity of skin symptoms and inflammation [32]. In this study, we showed improvement, not only in objective healing rates (more ulcers healed and faster healing rate), but also in subjective indicators, such as pain and health-related quality of life.

Chronic venous disease (CVD) and VLUs have many times been shown to be associated with poorer quality of life and impaired functioning in all areas—physical, mental, and social [15,16,17,18,19,20,21,55,56,57]. The negative impact of the disease increases with its advancement, and it is most severe at the stage of active ulcers [55,56,58]. The physical dimension, including pain reported at the beginning of therapy by as many as 95% of patients with VLUs [59], and during its duration, by 30–90% of patients [60], usually has the greatest impact on the quality of life. The mean pain intensity on a visual analog scale (0–11 points) was 5.86 [61]; more than half of the patients described it as “moderate to severe” [57], while others described it as “unbearable pain,” or “tear-inducing pain” [14]. In one study, the participants described the pain as unremitting, a constant reminder of the ulcer, and claimed it contributed to their feelings of loss of control [14,61]. Pain provoked discomfort, decreased the patients’ ability to take up physical activity and walk, and caused limitations in everyday life and work. In turn, at night, the pain disturbed effective sleep [16,17,20,55,57,62,63]. In our study, the pain dimension and related aspects, such as sleep quality, activity, and mobility, were also the most disrupted sphere of HRQoL. In addition, the Skindex-29 scale showed that patients experienced severe skin symptoms similar to dermatological conditions, such as soreness of the periwound skin, burning and stinging, tenderness, and itching. It is reported that these disorders are common [17,21,38,64], but effective therapy reduces their nuisance [65]. Jockenhöfer, F. et al. [66] report that even merely raising patients’ expectations for a novel treatment may have a significant impact on improving the quality of life associated with ulceration. Taking care of the patient, showing support, and showing the ways of coping with the ailments [20,65,66] increase patient’s satisfaction with the treatment. Therefore, in the study by Jockenhöfer, F. et al. [66], the placebo group and the intervention group did not differ significantly in terms of improvement in pain and quality of life.

We want to emphasize that we also observed healing progress, pain reduction, and improvement in quality of life in all patients. Both groups received the same standard of care, including compression therapy and wound bed care. However, the addition of an intervention in the form of collagen gel resulted in better results in both the healing process and the reduction in physical symptoms (mainly cutaneous). We only observed a reduction in the level of worry about the severity of the skin disease in the collagen group. The areas related to general physical fitness (crouching, kneeling down, walking at a brisk pace, playing a sport, exerting oneself physically) did not change significantly in any of the groups, which may be related to the age of the respondents, on average, 64 years. Studies evaluating other collagen products in the treatment of VLUs have also confirmed their benefits and showed improved quality of life [44,55,66,67,68]. Nevertheless, we are the first to apply collagen to the surrounding skin, showing an improvement in skin parameters.

## 5. Conclusions

This study showed that the application of fish collagen gel to the periwound skin improves the healing process and quality of life in patients with VLUs. The 12-week treatment with collagen reduced the severity of physical complaints, pain, and local skin symptoms, such as skin soreness, burning, stinging, and sensitivity, which determined the quality of life in patients with VLUs to the greatest extent.

## Figures and Tables

**Figure 1 ijerph-19-07108-f001:**
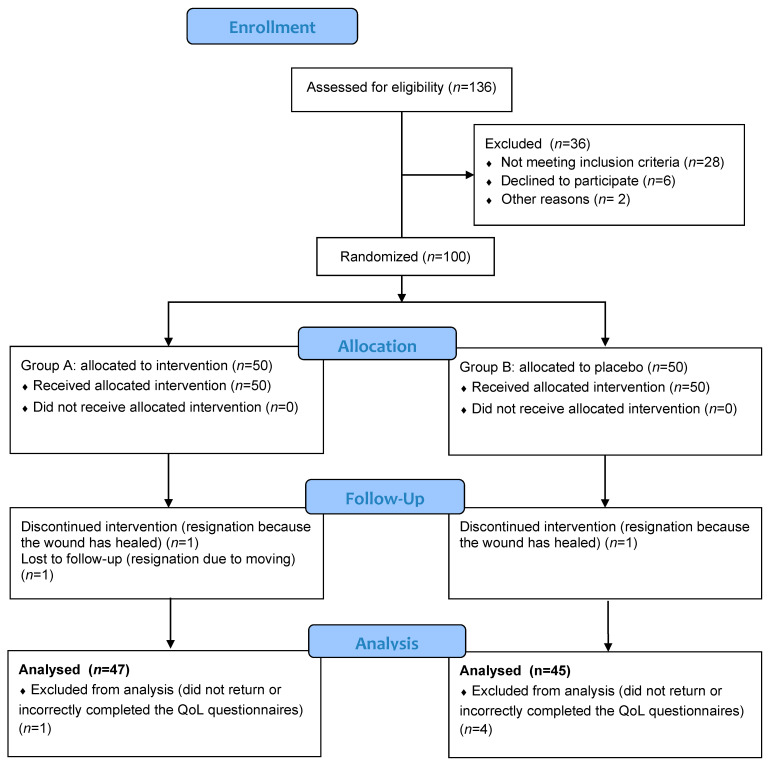
Flow diagram of the progress through the phases of a parallel randomized trial: A = tropocollagen group, B = placebo group.

**Figure 2 ijerph-19-07108-f002:**
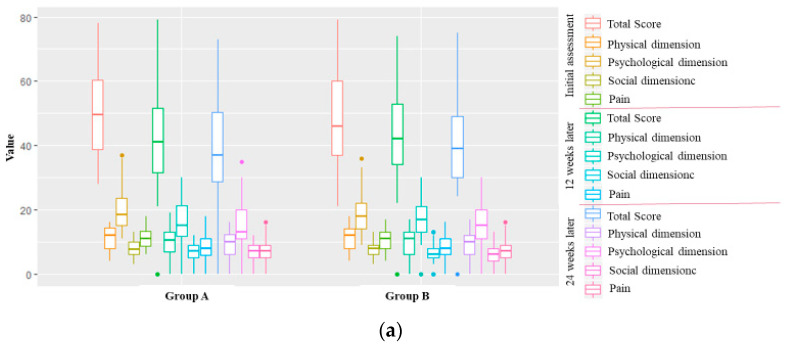
Box plots of changes in quality of life in the consecutive weeks of observation in the study groups. (**a**) Changes in quality of life according to the CIVIQ scale. Correlation between wound healing progress and changes in quality of life (total CIVIQ score): R (Group A) = 0.87, *p* = 0.000; R (Group B) = 0.85, *p* = 0.000. (**b**) Changes in quality of life according to the SKINDEX-19 scale. Correlation between wound healing progress and changes in quality of life (total Skindex-19 score): R (Group A) = 0.79, *p* = 0.001; R (Group B) = 0.76, *p* = 0.000. R—Spearman’s rank correlation coefficient; (A)—group A; (B)—group B; *p*—value.

**Figure 3 ijerph-19-07108-f003:**
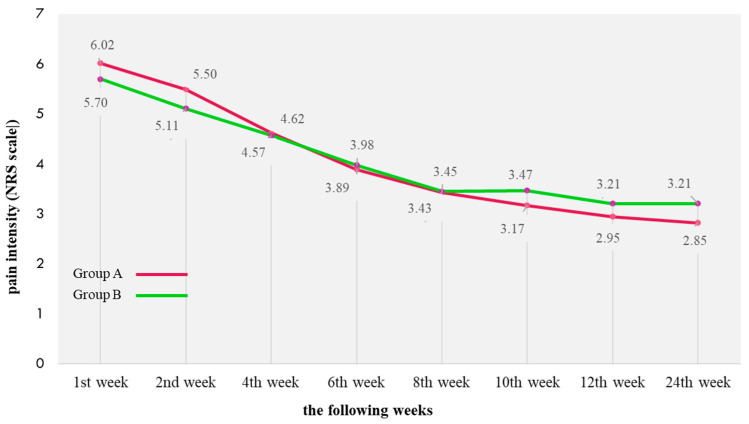
Changes in pain intensity on the NRS scale in the consecutive weeks of observation. Correlation between wound area and pain intensity (NRS scale): R_(Group A)_ = 0.152, *p* = 0.005; R_(Group B)_ = 0.29, *p* = 0.001. Correlation between wound healing progress and changes in pain intensity (NRS scale): R_(Group A)_ = 0.79, *p* = 0.001; R_(Group B)_ = 0.59, *p* = 0.001. R—Spearman’s rank correlation coefficient; (A)—group A; (B)—group B; *p*—value.

**Table 1 ijerph-19-07108-t001:** Characteristics of the patients included in the analysis.

Characteristic		Group A (*n* = 47)		Group B (*n* = 45)	
		Mean ± SD (Range)	Median	Mean ± SD (Range)	Median
Age (Years)		64.5 ± 12.00 (35–88)	64.0	63.6 ± 11.66 (39–87)	64.0
Gender *					
	Female	25 (53.2)		23 (48.9)	
	Male	22 (46.8)		24 (51.1)	
Duration of CVI (years)		17.7 ± 11.9 (1–50)	17.0	16.7 ± 13.7 (1–52)	13.0
Duration of VLU (months)		72.5 ± 102.5 (4–440)	20.0	40.8 ± 62.3 (2–360)	20.0
CEAP-C6 (score)		11.6 ± 1.63 (7–15)	12.0	11.6 ± 1.73 (7–16)	11.5
BMI		31.0 ± 7.2 (19.4–58,9)	30.7	30.8 ± 6.6 (20.6–58.9)	30.5
Initial wound size (cm^2^)		18.3 ± 15.1 (5–50)	11.5	15.4 ± 14.6 (5–50)	8.6
ABI right		1.09 ± 0.12 (0.90–1.33)	1.06	1.14 ± 0.14 (0.80–1.5)	1.14
ABI left		1.07 ± 0.12 (0.83–1.40)	1.06	1.11 ± 0.14 (0.72–1.5)	1.10

* Shown as the number of observations (percentage); SD—standard deviation, CVI—chronic venous insufficiency, VLU—venous leg ulcer, ABI—ankle-brachial pressure index.

**Table 2 ijerph-19-07108-t002:** Comparison of the quality of life and the ulcer healing process in the studied groups.

Domain	Group A, *n* = 47		Group B, *n* = 45		
	Mean ± SD	Median [IQR]	Mean ± SD	Median [IQR]	*p*
HR (cm^2^/week)12 weeks later	0.68 ± 0.61	0.56 [0.42, 0.78]	0.51 ± 0.68	0.43 [0.24, 0.68]	0.598
	13 (27.7%) *		11 (24.4%) *		0.675
HR (cm^2^/week) 24 weeks later	0.42 ± 0.44	0.33 [0.22, 0.49]	0.33 ± 0.31	0.24 [0.21, 0.41]	0.219
	25 (53.2%) *		18 (40.0%) *		0.223
**I. SKINDEX-29, Initial Assessment**					
PS	17.79 ± 5.13	17.00 [14.00, 21.00]	15.69 ± 5.21	16.00 [12.25, 21.75]	0.420
PF	23.96 ± 8.77	21.00 [18.00, 29.00]	21.73 ± 7.04	20.00 [16.00, 28.00]	0.497
ES	21.79 ± 7.35	21.00 [16.75, 25.00]	19.88 ± 5.60	20.00 [17.00, 25.00]	0.691
TS	63.53 ± 18.92	60.00 [49.50, 72.25]	57.31 ± 15.61	57.00 [47.00, 74.00]	0.479
**II. SKINDEX-29, 12 Weeks Later**					
PS	14.49 ± 4.49	14.00 [11.00, 17.00]	15.16 ± 5.09	16.00 [10.00, 19.00]	0.578
PF	19.32 ± 7.32	15.00 [12.75, 21.25]	17.51 ± 5.85	16.00 [12.00, 21.00]	0.968
ES	18.98 ± 7.08	14.00 [11.00, 18.25]	16.18 ± 4.83	15.00 [11.00, 19.00]	0.651
TS	52.79 ± 17.34	47.00 [39.00, 56.25]	52.87 ± 14.93	50.00 [38.00, 64.00]	0.516
**III. SKINDEX-29, 24 Weeks Later**					
PS	13.47 ± 5.12	11.50 [9.00, 15.25]	13.29 ± 4.57	13.00 [9.00, 15.00]	0.951
PF	18.30 ± 8.05	14.00 [11.00, 18.50]	16.16 ± 4.98	14.00 [11.00, 18.00]	0.951
ES	18.23 ± 7.06	12.50 [10.00, 20.00]	15.00 ± 4.73	14.00 [11.00, 19.00]	0.811
TS	50.00 ± 18.53	41.50 [35.00, 54.75]	48.40 ± 13.91	43.00 [35.00, 55.00]	0.673
**CIVIQ, Initial Assessment**					
Pain	11.13 ± 3.30	11.00 [8.75, 13.25]	10.80 ± 3.47	11.00 [8.00, 13.00]	0.533
PhD	11.28 ± 3.66	12.00 [8.00, 14.25]	11.24 ± 3.61	12.00 [8.00, 14.00]	0.811
PsD	20.13 ± 6.83	18.50 [15.00, 23.50]	18.84 ± 6.46	8.00 [6.00, 9.00]	0.437
ScD	7.89 ± 2.80	7.50 [6.00, 10.00]	7.87 ± 2.74	8.00 [6.00, 9.00]	0.749
TS	50.43 ± 13.40	50.00 [38.50, 60.50]	48.76 ± 13.79	50.00 [38.50, 60.00]	0.444
GIS	38.03 ± 16.75	37.50 [23.12, 50.62]	35.95 ± 17.01	38.75 [22.50, 50.000]	
**CIVIQ, 12 Weeks Later**					
Pain	8.74 ± 3.54	8.00 [6.00, 11.00]	8.78 ± 3.32	7.00 [6.00, 11.00]	0.667
PhD	10.28 ± 3.28	10.50 [6.75, 13.00]	10.47 ± 3.92	11.00 [6.00, 13.00]	0.854
PsD	16.94 ± 6.02	15.00 [11.75, 21.25]	17.67 ± 5.40	17.00 [13.00, 21.00]	0.548
ScD	7.30 ± 2.67	7.00 [5.00, 9.00]	7.02 ± 2.46	6.00 [5.00, 8.00]	0.358
TS	43.26 ± 13.52	41.00 [31.50, 51.50]	43.93 ± 12.11	42.00 [34.00, 53.00]	0.860
GIS	29.06 ± 16.90	26.25 [16.87, 40.00]	29.91 ± 14.96	28.75 [20.00, 42.50]	
**CIVIQ, 24 Weeks Later**					
Pain	8.00 ± 3.21	7.00 [5.00, 9.00]	7.87 ± 3.69	7.00 [5.00, 9.00]	0.816
PhD	10.15 ± 3.62	10.00 [6.00, 12.25]	10.24 ± 4.87	10.00 [6.00, 12.00]	0.845
PsD	16.44 ± 6.31	13.00 [11.00, 20.00]	16.89 ± 3.18	15.00 [11.00, 20.00]	0.685
ScD	7.13 ± 2.57	7.00 [5.00, 9.00]	6.96 ± 2.59	6.00 [4.00, 8.00]	0.279
TS	41.04 ± 13.54	37.00 [28.75, 50.25]	41.96 ± 11.97	39.00 [30.00, 49.00]	0.877
GIS	26.30 ± 17.03	21.25 [12.50, 38.12]	27.45 ± 18.13	25.00 [15.00, 37.50]	

SD—standard deviation, IQR—interquartile range, *p*—value (nonparametric Kruskal–Wallis test), SMD—standardized mean difference, HR—ulcer healing rate [cm^2^/week], * the number (and percent) of completely healed ulcers, PS—physical symptoms, PF—psychosocial functioning, ES—emotional sphere, TS—Total score, PhD—physical dimension, PsD—psychological dimension, ScD—social dimension, GIS—Global Index Score.

**Table 3 ijerph-19-07108-t003:** Comparison of changes in the quality of life in individual items of the Skindex-29 scale in the studied groups.

Item		Group A					Group B				
		Mean ± SD	Mean ± SD	Mean ± SD	I–II	I–III	Mean ± SD	Mean ± SD	Mean ± SD	I–II	I–III
1	My skin hurts **	2.47 ± 1.40	2.11 ± 1.20	1.70 ± 1.00	0.36	0.77	2.40 ± 1.34	2.24 ± 1.25	1.89 ± 1.15	0.16	0.51
2	My skin condition affects how well I sleep **	2.43 ± 1.30	1.85 ± 1.08	1.68 ± 1.07	0.57	0.74	2.47 ± 1.39	2.13 ± 1.20	1.87 ± 1.04	0.33	0.60
3	I worry that my skin condition may be serious **	3.26 ± 1.19	2.83 ± 1.32	2.74 ± 1.28.	0.43	0.51	3.00 ± 1.07	2.89 ± 1.07	2.82 ± 1.03	0.11	0.18
4	My skin condition makes it hard to work or do hobbies *	2.47 ± 1.27	2.00 ± 1.12	1.74 ± 0.99	0.47	0.72	2.51 ± 1.25	2.18 ± 1.07	2.09 ± 1.04	0.33	0.42
5	My skin condition affects my social life *	2.22 ± 1.23	1.62 ± 0.95	1.55 ± 0.93	0.60	0.66	1.82 ± 1.03	1.51 ± 0.76	1.31 ± 0.60	0.31	0.51
6	My skin condition makes me feel depressed *	2.51 ± 1.16	2.09 ± 1.10	2.02 ± 1.13	0.43	0.49	2.44 ± 1.22	1.80 ± 0.92	1.84 ± 0.85	0.64	0.60
7	My skin condition burns or strings **	2.83 ± 1.22	2.43 ± 1.17	2.02 ± 1.15	0.40	0.81	2.56 ± 1.20	2.27 ± 1.23	2.09 ± 1.02	0.29	0.47
8	I tend to stay at home because of my skin condition	2.11 ± 1.17	1.55 ± 0.97	1.53 ± 0.95	0.55	0.57	2.09 ± 1.20	1.56 ± 0.92	1.42 ± 0.81	0.53	0.62
9	I worry about getting scars from my skin condition **	1.68 ± 1.04	1.60 ± 1.10	1.40 ± 0.74	0.08	0.28	1.47 ± 0.76	1.76 ± 1.03	1,58 ± 0.87	-0.29	-0.11
10	My skin itches	3.00 ± 1.25	2.72 ± 1.23	2.53 ± 1.27	0.28	0.47	3.04 ± 1.30	2.71 ± 1.10	2.50 ± 1.11	0.33	0.54
11	My skin condition affects how close I can be with those I love	1.68 ± 0.91	1.51 ± 0.86	1.34 ± 0.73	0.17	0.34	1,58 ± 1.06	1.36 ± 0.68	1.33 ± 0.67	0.22	0.25
12	I am ashamed of my skin condition *	2.02 ± 1.15	1.98 ± 1.34	1.70 ± 1.02	0.04	0.32	1.87 ± 1.08	1.93 ± 1.01	1.78 ± 0.97	-0.07	0.09
13	I worry that my skin condition may get worse	2.77 ± 1.24	2.68 ± 1.14	2.72 ± 1.14	0.09	0.04	2.69 ± 0.95	2.78 ± 1.22	2.67 ± 1.22	-0.09	0.02
14	I tend to do things by myself because of my skin condition **	2.17 ± 1.19	1.89 ± 1.20	1.74 ± 1.15	0.28	0.43	2.27 ± 1.25	1.67 ± 1.13	1.47 ± 0.97	0.60	0.80
15	I am angry about my skin condition	1.81 ± 1.06	1.51 ± 0.88	1.49 ± 0.78	0.30	0.32	1.87 ± 1.12	1.47 ± 0.84	1.42 ± 0.81	0.40	0.44
16	Water bothers my skin condition (bathing, washing) **	1.85 ± 1.18	1.30 ± 0.69	1.40 ± 0.80	0.55	0.45	1.49 ± 0.89	1.42 ± 0.94	1.29 ± 0.73	0.07	0.20
17	My skin condition makes showing affection difficult	1.74 ± 1.11	1.47 ± 0.95	1.40 ± 0.77	0.28	0.34	1.53 ± 0.99	1.31 ± 0.76	1.24 ± 0.61	0.22	0.29
18	My skin is irritated	2.62 ± 1.23	2.06 ± 0.96	2.15 ± 1.06	0.55	0.47	2.73 ± 1.23	2.29 ± 1.18	2.09 ± 1.18	0.44	0.64
19	My skin condition affects my interactions with others	1.51 ± 0.88	1.23 ± 0.56	1.36 ± 0.76	0.28	0.15	1.51 ± 1.01	1.36 ± 0.74	1.33 ± 0.71	0.16	0.18
20	I am embarrassed by my skin condition *	2.13 ± 1.10	1.60 ± 0.92	1.60 ± 0.85	0.53	0.53	2.16 ± 1.13	1.84 ± 1.00	1.71 ± 0.92	0.31	0.44
21	My skin condition is a problem for the people I love	1.87 ± 1.15	1.53 ± 0.95	1.43 ± 0.90	0.34	0.45	1.93 ± 1.23	1.31 ± 0.68	1.36 ± 0.74	0.62	0.58
22	I am frustrated by my skin condition	1.81 ± 1.04	1.74 ± 0.99	1.68 ± 0.96	0.06	0.13	1.96 ± 1.11	1.62 ± 0.86	1.53 ± 0.84	0.33	0.42
23	My skin is sensitive **	2.98 ± 1.11	2.38 ± 0.99	2.28 ± 1.02	0.60	0.70	2.91 ± 1.29	2.80 ± 1.16	2.47 ± 1.08	0.11	0.44
24	My skin condition affects my desire to be with people	1.68 ± 1.02	1.45 ± 0.75	1.34 ± 0.73	0.23	0.34	1.60 ± 1.10	1.29 ± 0.59	1.27 ± 0.62	0.31	0.33
25	I am humiliated by my skin condition	1.66 ± 0.96	1.36 ± 0.76	1.34 ± 0.64	0.30	0.32	1.64 ± 1.05	1.36 ± 0.74	1.29 ± 0.69	0.29	0.36
26	My skin condition bleeds	2.04 ± 1.04	1.49 ± 0.80	1.38 ± 0.80	0.55	0.66	1.82 ± 1.07	1.42 ± 0.72	1.38 ± 0.68	0.40	0.44
27	I am annoyed by my skin condition	2.15 ± 1.23	1.60 ± 0.92	1.53 ± 0.86	0.55	0.62	1.96 ± 1.04	1.51 ± 0.84	1.49 ± 0.84	0.44	0.47
28	My skin condition interferes with my sex life	1.47 ± 1.00	1.26 ± 0.87	1.11 ± 0.60	0.21	0.36	1.62 ± 1.17	1.24 ± 0.68	1.24 ± 0.83	0.38	0.38
29	My skin condition makes me tired	2.62 ± 1.13	1.96 ± 1.08	2.06 ± 1.15	0.66	0.55	2.40 ± 1.12	1.84 ± 1.07	1.80 ± 1.06	0.56	0.60

Mean—average point value of each item, SD—standard deviation, * items in which changes in the average quality of life differed significantly (*p* < 0.05) between the groups in at least one of the analyzed periods (I–II or I–III), ** items in which changes in the average quality of life differed significantly (*p* < 0.05) between the groups in both analyzed periods (I–II and I–III).

**Table 4 ijerph-19-07108-t004:** Comparison of changes in the quality of life in individual statements of the CIVIQ scale in the studied groups.

Statement		Group A					Group B				
		Mean ± SD	Mean ± SD	Mean ± SD	I–II	I–III	Mean ± SD	Mean ± SD	Mean ± SD	I–II	I–III
1	Have you had any pain in your ankles or legs, and how severe has this pain been? *	3.17 ± 1.13	2.36 ± 1.26	1.72 ± 0.97	0.81	1.45	2.93 ± 1.21	2.20 ± 1.08	1.91 ± 1.14	0.73	1.02
2	How much trouble have you experienced at work or during your usual daily activities because of your leg problems? *	2.77 ± 0.96	2.17 ± 1.03	2.09 ± 0.97	0.60	0.68	2.49 ± 1.10	2.24 ± 0.98	1.93 ± 0.96	0.24	0.56
3	How much trouble have you experienced at work or during your usual daily activities because of your leg problems?	2.51 ± 1.28	1.79 ± 1.14	1.53 ± 1.02	0.72	0.98	2.71 ± 1.22	1.93 ± 1.05	1.62 ± 1.01	0.78	1.09
4	How much trouble have you had remaining standing for a long time? *	2.68 ± 1.12	1.43 ± 1.21	2.26 ± 1.19	0.26	0.43	2.67 ± 1.26	2.40 ± 1.10	2.40 ± 1.18	0.27	0.27
5	Climbing several flights of stairs?	2.74 ± 1.15	2.28 ± 1.14	2.45 ± 1.08	0.47	0.30	2.51 ± 1.12	2.36 ± 1.15	2.27 ± 1.05	0.16	0.24
6	Crouching, kneeling down	3.38 ± 1.17	3.23 ± 1.24	2.94 ± 1.19	0.15	0.45	3.47 ± 1.22	3.29 ± 1.34	3.27 ± 1.30	0.18	0.20
7	Walking at a brisk pace	2.98 ± 1.19	2.94 ± 1.24	2.79 ± 1.20	0.04	0.19	3.16 ± 1.22	2.93 ± 1.27	2.87 ± 1.20	0.22	0.29
8	Travelling by car, bus, plane	2.15 ± 1.12	1.85 ± 1.00	1.85 ± 1.02	0.30	0.30	2.22 ± 1.04	1.73 ± 1.03	1.76 ± 0.93	0.49	0.47
9	Performing household tasks (e.g., standing and moving around in the kitchen, carrying a child in your arms, ironing, cleaning the floor or dusting the furniture) *	2.17 ± 1.05	1.83 ± 0.96	1.74 ± 0.90	0.34	0.43	2.11 ± 0.98	1.89 ± 0.91	1.84 ± 0.95	0.22	0.27
10	Going out for the evening, going to a wedding, a party, a cocktail party	2.32 ± 1.16	1.96 ± 1.04	1.94 ± 1.01	0.36	0.38	2.21 ± 1.14	1.73 ± 0.99	1.71 ± 0.99	0.47	0.49
11	Playing a sport, exerting yourself physically	3.43 ± 1.17	3.49 ± 1.28	3.42 ± 1.36	−0.06	0.00	3.44 ± 1.34	3.56 ± 1.22	3.49 ± 1.29	−0.11	−0.04
12	I have felt nervous/tense *	2.15 ± 1.06	1.85 ± 1.00	1.55 ± 0.95	0.30	0.60	2.04 ± 1.11	1.77 ± 1.10	1.08 ± 1.04	0.27	0.24
13	I have become tired quickly **	2.66 ± 1.09	2.19 ± 1.06	1.83 ± 1.07	0.47	0.83	2.58 ± 1.01	2.44 ± 1.12	2.20 ± 1.01	0.13	0.38
14	I have felt I am a burden	1.62 ± 0.99	1.45 ± 0.85	1.30 ± 0.75	0.17	0.32	1.58 ± 0.87	1.36 ± 0.71	1.24 ± 0.57	0.22	0.33
15	I have had to be cautious all the time	2.79 ± 1.12	2.36 ± 1.07	2.43 ± 1.04	0.43	0.36	2.87 ± 1.25	2.73 ± 1.25	2.93 ± 1.05	0.13	−0.07
16	I have felt embarrassed about showing my legs **	2.79 ± 1.37	2.34 ± 1.27	2.17 ± 1.19	0.45	0.62	2.16 ± 1.19	2.24 ± 1.30	2.13 ± 1.10	−0.09	0.02
17	I have become irritated easily **	2.68 ± 1.32	2.17 ± 1.29	1.98 ± 1.21	0.51	0.70	2.33 ± 1.13	2.31 ± 1.12	2.24 ± 1.09	0.02	0.09
18	I have felt as if I am handicapped *	1.98 ± 1.24	1.64 ± 1.05	1.62 ± 0.99	0.34	0.36	1.82 ± 1.21	1.71 ± 1.12	1.60 ± 0.91	0.11	0.22
19	I have found it hard to get going in the morning	1.66 ± 1.20	1.40 ± 0.77	1.70 ± 1.00	0.26	0.04	1.69 ± 1.06	1.67 ± 1.00	1.44 ± 0.84	0.02	0.24
20	I have not felt like going out	1.81 ± 1.04	1.57 ± 0.96	1.64 ± 1.03	0.24	0.17	1.78 ± 1.06	1.47 ± 0.81	1.29 ± 0.63	0.31	0.49

Mean—average point value of each statement, SD—standard deviation, * items in which changes in the average quality of life differed significantly (*p* < 0.05) between the groups in at least one of the analyzed periods (I–II or I–III), ** items in which changes in the average quality of life differed significantly (*p* < 0.05) between the groups in both analyzed periods (I–II and I–III).

## Data Availability

Not applicable.

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
