# Peer review of "Healing Process, Pain, and Health-Related Quality of Life in Patients with Venous Leg Ulcers Treated with Fish Collagen Gel: A 12-Week Randomized Single-Center Study"

_ijerph, 2022, doi:10.3390/ijerph19127108_

Round 1

Reviewer 1 Report

I thought it was very good.  there are some minor spelling error.   I am curious why you did not apply the fish collagen to the wound bed itself??  The collagen in the wound is being degraded by the MMP's and other proteases and it needs a scaffold applies as well as collagen to absorb and bind to the proteases to sort of use the up. why the periowound skin to apply the gel? I thought it was very interesting. please advise.

Author Response

Dear Reviewer,

kind regards,

Paulina Mościcka

Reviewer 2 Report

Although the authors suggested that they are the first to apply fish collagen gel to the surrounding skin and show improvement in skin parameters,  the authors have already confirmed the effect of fish collagen gel in Reference. 

Further, they discuss the relationship with patients' quality of life in this study, but no significant difference was observed between the placebo group except for physical complaints. Therefore, we have no choice but to reject the current paper because it is very similar to the previous paper and lacks novelty.

Author Response

(The authors gave the same response as above.)

Author Response

(The authors gave the same response as above.)

Round 2

Reviewer 2 Report

Authors thoughtfully revised manuscript and addressed reviewer comments by adding additional text, tables and figures. I think this manuscript is suitable for publication in IJERPH.

Reviewer 3 Report

I agree with the changes made by the authors. I recommend Accept the manuscript.